# Prion–like Proteins in Plants: Key Regulators of Development and Environmental Adaptation via Phase Separation

**DOI:** 10.3390/plants13182666

**Published:** 2024-09-23

**Authors:** Peisong Wu, Yihao Li

**Affiliations:** 1Faculty of Arts and Sciences, Beijing Normal University, Zhuhai 519087, China; peisongwu@mail.bnu.edu.cn; 2Center for Biological Science and Technology, Guangdong Zhuhai–Macao Joint Biotech Laboratory, Advanced Institute of Natural Science, Beijing Normal University, Zhuhai 519087, China

**Keywords:** prion–like proteins, liquid–liquid phase separation, prion–like domains, plant development, stress responses

## Abstract

Prion–like domains (PrLDs), a unique type of low–complexity domain (LCD) or intrinsically disordered region (IDR), have been shown to mediate protein liquid–liquid phase separation (LLPS). Recent research has increasingly focused on how prion–like proteins (PrLPs) regulate plant growth, development, and stress responses. This review provides a comprehensive overview of plant PrLPs. We analyze the structural features of PrLPs and the mechanisms by which PrLPs undergo LLPS. Through gene ontology (GO) analysis, we highlight the diverse molecular functions of PrLPs and explore how PrLPs influence plant development and stress responses via phase separation. Finally, we address unresolved questions about PrLP regulatory mechanisms, offering prospects for future research.

## 1. Introduction

Prions, derived from the term “proteinaceous infectious particles”, represent a distinct class of proteins found predominantly in mammals [1,2]. Prions exist in dual conformational states, which are fundamental to their biological functions and roles in disease pathology. The normal cellular form, known as PrP^C^, performs vital neurophysiological functions, while the misfolded and pathogenic isoform, PrP^Sc^, is characterized by its infectivity [3,4,5]. PrP^Sc^ has self–replicating capabilities, triggering the conversion of PrP^C^ into its pathogenic form [3,4,5]. This conversion leads to the aggregation of misfolded proteins into amyloid deposits, resulting in a cascade of neurodegenerative diseases [2,6,7]. Prion–like proteins (PrLPs), which contain prion–like domains (PrLDs) with structural and amino acid composition similarities to prions [8,9,10,11,12], are found across a spectrum of organisms, including animals, yeasts, and plants [2,13,14,15].

Although it used to be considered that PrLPs are associated with diseases, their prion–like properties are now recognized as playing a pivotal role in cellular physiology, particularly their ability to undergo phase separation. Phase separation is a cellular physiological phenomenon involving the concentration of proteins and other biomacromolecules into membrane–less organelles (MLOs) distinct from the surrounding cellular environment [16,17]. This process orchestrates the spatial arrangement and regulation of cellular biochemistry, allowing the precise modulation of biochemical reactions independent of membrane–bound organelles [17,18]. Protein liquid–liquid phase separation (LLPS), leading to the formation of condensates, is common in cells and generally mediated by multivalence interactions involving proteins that contain low–complexity domains (LCDs) or intrinsically disordered regions (IDRs) [19,20]. Recent studies indicate that PrLDs, a unique type of LCD or IDR, are crucial for driving protein phase separation [21,22,23]. In this review, we elucidate the mechanisms underlying the phase separation of PrLPs and their implications for plant developmental processes and adaptation to environmental stresses.

## 2. Structural Features of PrLPs and Mechanisms in Forming Phase Separation

PrLPs typically comprise one or more PrLDs along with functional domains. PrLDs are often enriched in polar amino acids such as glutamine and asparagine, which facilitate β–sheet formation and amyloid fibril nucleation [2,7,23]. However, only a few PrLPs have been shown to form amyloid fibrils [11,24]. In addition to glutamine and asparagine, other polar amino acids, including serine and tyrosine, are also enriched in PrLDs, while hydrophobic amino acids are generally lacking [23]. This unique amino acid composition gives PrLPs low sequence complexity and a largely unfolded state, contributing to their prion–like behavior [23]. Studies indicate that the prion–like behavior of these domains is determined by their amino acid compositions rather than by their specific amino acid sequence [25,26]. The identification of PrLPs has been facilitated by tools such as the PLAAC (Prion–Like Amino Acid Composition) algorithm, based on a hidden Markov model (HMM) trained on known yeast prion sequences [27]. The application of this tool has revealed that PrLPs not only exist in yeast and mammalian genomes but are also extensively present across the plant kingdom [14,27].

One key feature of PrLPs is their capacity to mediate phase separation through multivalent interactions, which occur either between proteins or between proteins and nucleic acids and are crucial for phase separation [20,28]. PrLPs can mediate intramolecular or intermolecular interactions, promoting homotypic phase separation [29]. Additionally, PrLPs can also form scaffolds to recruit other proteins, leading to condensate formation, a process known as heterotypic phase separation [29,30]. Based on research into prion–like RNA–binding proteins (PrL–RBPs), a “stickers–and–spacers” model has been proposed to elucidate phase separation mediated by PrLPs [23,31,32,33]. In this model, specific physicochemical properties of the amino acids modulate the phase behavior and material characteristics of the biomolecular condensates [23,31]. Stickers are motifs that mediate cohesive interactions, forming reversible and non–covalent crosslinks [32,34]. Specifically, aromatic residues such as tyrosine and phenylalanine, along with charged residues such as arginine and lysine within RNA–binding domains (RBDs), are identified as primary stickers [23,32]. These residues engage in π–π and cation–π interactions, which are critical for forming the crosslinks that drive phase separation [23,35,36,37]. Among these, tyrosine is a more potent sticker than phenylalanine [23]. Arginine functions as an auxiliary binder, with its effectiveness influenced by the environmental context [37]. Conversely, lysine reduces the interactions between stickers [23,37]. Spacers provide a structural scaffold for stickers and are crucial in determining the effective solvation volume (*v_es_*) [32,34]. The *v_es_* is calculated as the average volume per residue allocated for interactions with the solvent, affecting the solubility and phase behavior of the protein [32]. Spacer compositions, typically glycine, glutamine, and serine, play significant roles [23]. Glycine maintains the liquidity of the condensates, while glutamine and serine promote condensate hardening [23]. The distribution of stickers and the positioning of spacers can influence the characteristics of condensates [33].

Furthermore, RNA plays a regulatory role in the phase separation of PrLPs, particularly in PrL–RBPs where RNA–binding domains (RBDs) are prevalent [38]. Structured RNAs can act as scaffolds within condensates, facilitating nucleation and driving condensate formation [38,39]. For instance, stress granules (SGs) are rich in structured mRNAs, which potentially serve as nucleation scaffolds for SG formation [39]. Low RNA concentrations can promote condensate formation, whereas high RNA concentrations or high RNA/protein ratios can inhibit condensate assembly [39]. The negatively charged RNA can interact with cationic peptides upon reaching a state of charge neutralization. These multivalent interactions facilitate the formation of liquid–like condensates [40,41].

## 3. The Molecular Function of PrLP Condensates

Due to the diverse functional domains within PrLPs, these proteins have been shown to exhibit various regulatory roles in physiological and biochemical processes in yeast and animal cells. To understand the function of PrLPs in plants, Gene Ontology (GO) analysis of PrLP genes across ten common monocot and dicot plant species reveals that many of their molecular functions are conserved (Figure 1). This analysis indicates that PrLPs are enriched in nucleic acid binding and nuclear compartments, where they often function as RBPs involved in gene transcription and RNA processing, consistent with their roles in yeast and mammalian cells [21]. Additionally, they are also enriched in membrane–less, nucleic acid–rich organelles such as the RISC complex, ribonucleoprotein (RNP) granules, and processing bodies (P–bodies, PBs), which have been shown to form by phase separation. Moreover, PrLPs are implicated in vesicle–mediated transport and vesicle budding from membranes. Correspondingly, they are found in intracellular membrane–bounded organelles and clathrin–coated vesicles, indicating their widespread role in the regulation of membrane dynamics (Appendix B).

### 3.1. Transcriptional Regulation

GO analysis reveals that numerous PrLP genes encode transcription factors or transcriptional regulators. These factors can form condensates under specific conditions, which may have contrasting effects on gene transcription. On the one hand, the condensate can sequester transcription factors and/or disrupt transcriptional regulatory complex formation, thereby compromising its functionality [42]. On the other hand, some PrLPs form condensates that concentrate transcription–related proteins, either enhancing or sustaining the transcription of downstream genes [43,44,45,46]. Furthermore, the LLPS of specific PrLPs can localize to specific gene loci and facilitate the formation of specialized transcriptional regulatory structures such as R−loops [47]. The R−loop structure comprises RNA–DNA hybrids and single−stranded DNA [48]. HRLP (hnRNP R−LIKE PROTEIN), along with splicing factor SR45 (ARGININE/SERINE−RICH45), forms condensates that enhance R−loop formation, which in turn affects the recruitment of transcription−related proteins such as RNA polymerase II (RNA Pol II) and impedes co−transcriptional splicing of the pre−mRNA located within the R−loop [47,48]. The intricate interplay between PrLPs and their LLPS−driven condensates highlights the sophisticated regulation of gene expression in response to various cellular signals [46,47,49].

### 3.2. RNA Processing

Numerous PrLPs possess unique RBDs that enable their interaction with RNA molecules [50]. These domains include YTH (YT521−B homology) [51], RRM (RNA recognition motif) [52], DEAD−box [53], KH (K homology) [54], and ALBA (acetylation lowers binding affinity) domains [55]. These PrL−RBPs can undergo LLPS, either independently or by recruiting other RNA−processing factors, to form condensates. The condensates gather RNA and other RNA−processing factors into membrane−less cellular compartments, which enhance the efficiency and precision of RNA−processing events, including alternative splicing [54,56] and polyadenylation [51,52]. This coordination contributes to the precise regulation of gene expression.

RNA metabolic processes and mRNA processing are enriched in GO terms across various plant species. PrL−RBPs can undergo phase separation with mRNA, thereby participating in mRNA processing [23,39]. PrL−RBPs with RRM domains, such as FCA (FLOWERING CONTROL LOCUS A) and FPA (FLOWERING LOCUS PA), form condensates that participate in mRNA 3′ proximal site selection and polyadenylation processing [52]. Additionally, PrLPs also contribute to the biogenesis and maturation of small noncoding RNAs. These nuclear RNP condensates, known as dicing bodies (D−bodies), small interfering RNA (siRNA) bodies, and Cajal bodies, facilitate different stages of RNA processing. D−bodies convert pri−miRNA (primary microRNA) into pre−miRNA (precursor microRNA), while siRNA bodies and Cajal bodies produce siRNA. Cajal bodies are also involved in the processing of snRNA (small nuclear RNA) [53,57,58,59]. PrL−RBPs such as DEAD−box RH (RNA helicase) 8/12 facilitate the formation of D−bodies and siRNA bodies, while NRPD1b recruits siRNA in Cajal bodies [53,58,59].

### 3.3. Formation of Cytoplasmic RNP Granules

GO analysis indicates that PrLPs play a crucial role in the assembly of RNP granules in the cytoplasm, such as SGs and PBs, which are essential for mRNA regulation. SGs are dynamic MLOs that rapidly assemble under stress conditions [60,61,62]. SGs participate in the post−transcriptional regulation and translational processes of mRNAs, particularly those encoding stress−responsive genes [63]. mRNAs can be temporarily stored in these condensates, where their translation is suppressed. PrL−RBPs with YTH domains, such as ECT1 (EVOLUTIONARILY CONSERVED C−TERMINAL REGION 1) and ECT8, act as N^6^−methyladenosine (m^6^A) readers, binding to m^6^A−modified RNA and forming condensates to sequester these m^6^A−modified mRNAs, serving as a protective mechanism to stress [64,65,66,67,68].

PBs are another type of cytoplasmic RNP granules that exhibit properties of liquid−like condensates. They function as sites for RNA sequestration and degradation [64,69,70,71]. Key mRNA decapping factors, including DCP1 (DECAPPING1), DCP2, and DCP5, are integral components of PBs. Notably, DCP1 and DCP5 have been identified as PrLPs [45,72,73]. The assembly and disassembly of PBs are linked to the activation of plant immune response, where mRNAs of immune−suppressive genes are degraded [74]. SGs and PBs frequently share several components, suggesting potential for protein and RNA exchange between these organelles, as supported by recent research [75,76]. For instance, PrL−RBPs RH6/8/12 are present in both SGs and PBs. The *rh6/rh8/rh12* triple mutant shows defects in the formation of PBs and SGs [77]. Additionally, PBs share common components with siRNA bodies, a type of RNP granule involved in the biogenesis of siRNA [59,78].

### 3.4. Membrane−Related Regulation

GO analysis suggests a potential link between PrLPs and the regulation of membrane behaviors, such as vesicle−mediated transport and vesicle budding. Although the involvement of PrLPs in these processes remains poorly understood in plants, other IDPs have been shown to participate through LLPS. During endocytosis and vesicle formation, the LLPS of IDPs can induce membrane invagination, which enhances membrane curvature [79,80]. Membranes also serve as scaffolds for the assembly or disassembly of condensates [73,81,82]. The PB component DCP1, a PrLP, can interact with the SCAR (the suppressor of the cAMP receptor)–WAVE (WASP family verprolin homolog) complex at cell edges and vertices. DCP1 undergoes LLPS with the SCAR–WASP complex, promoting PB disassembly and actin nucleation on the plasma membrane [73]. Additionally, condensate formation facilitates the enveloping of membrane structures, which is beneficial for processes such as substance transport and the formation of autophagosomes [82,83]. Further research is warranted to elucidate the membrane−associated functions of PrLPs in plants.

## 4. Functional Roles of PrLPs in Plant Development

Throughout the plant life cycle, from seed germination to flowering and fruiting, various PrLPs are pivotal in modulating developmental pathways. These proteins also enhance the plant’s ability to perceive environmental changes, thereby facilitating the rapid adaptation of developmental strategies.

### 4.1. Meristem Maintenance

Plant growth relies on the meristem [84,85]. The root apical meristem (RAM) and shoot apical meristem (SAM) are regions containing pluripotent stem cells that maintain the stem cell niches (SCNs), enabling differentiation into root and shoot tissues [86,87]. Precise control of gene expression by key transcription factors is essential for maintaining stem cell activity in these meristem regions. Recent studies indicate that PrLD−containing transcription factors are crucial for RAM and SAM maintenance by modulating transcriptional regulation through promoting LLPS [43,88].

In the RAM, the SCN consists of quiescent center (QC) cells, which are essential for maintaining surrounding stem cells in a non−autonomous manner. This maintenance ensures the continuous production of new cells for root growth and tissue differentiation [86,87,89,90]. The PLT (PLETHORA) and the WOX (WUSCHEL−RELATED HOMOLOGY BOX) transcription factor families are integral to this regulatory process [91,92]. WOX5 and the PLTs, including PLT1/2/3/4, are co–expressed in the SCN and work in concert to preserve the stem cell identity of the QC [88,92,93,94]. The translocation of WOX5 to the columella stem cells (CSCs) results in its incorporation into nuclear bodies (NBs) by PLTs, thereby inhibiting its activity. Each PLT has PrLDs at its C−terminus, endowing it with the ability to undergo LLPS and form NBs. These NBs sequester WOX5 and RNA, crucial for sustaining the stem cell properties of CSCs and guiding their developmental fate [88] (Figure 2B).

The maintenance of the SAM relies on KNOX1 (CLASS 1 KNOTTED1−LIKE HOMEOBOX) transcription factors, which are crucial for preventing cell differentiation and sustaining the meristematic state [95]. In *Arabidopsis*, four KNOX1 proteins, including STM (SHOOT MERISTEMLESS), play an important role in SAM initiation and maintenance [43,96,97]. STM contains a PrLD and forms nuclear condensates, a process that is facilitated by PrLD−containing BELL (BEL1−like) proteins. Additionally, the Mediator complex subunit MED8, which contains a PrLD and is part of the transcriptional machinery, is associated with the STM condensates. Condensate formation enhances the transcriptional regulatory activity of STM and strengthens its ability to maintain the meristem. STM exhibits significantly stronger meristematic activity than the other three KNOX1 proteins, which lack PrLDs and do not undergo phase separation. This difference highlights the importance of PrLD in STM condensate formation and meristematic activity [43] (Figure 2A).

### 4.2. Light Signaling Regulation

Light is a crucial environmental signal, serving both as an essential factor for photosynthesis and as a cue for plant photomorphogenesis. During photosynthesis, the MLO pyrenoids increase the local concentration of CO_2_ and promote the aggregation of rubisco [98]. In *Chlamydomonas reinhardtii*, the EPYC1 (essential pyrenoid component 1), featuring IDRs, interacts with rubisco and undergoes LLPS to form pyrenoids [99,100,101,102]. Similarly, PrLP PYCO1 (pyrenoid component 1) typically interacts with rubisco in microalgae [30]. PYCO1 undergoes LLPS via its sticker motifs, particularly in the R6 region rather than the PrLD. PYCO1 forms pyrenoids that offer a structural scaffold for rubisco, thereby enhancing its CO_2_ fixation efficiency [30] (Figure 2C).

Plant cells discern various light qualities via a spectrum of photoreceptors, modulating plant growth and development [103]. Recent findings indicate that red and far−red light act as a key regulatory switch for the alternative splicing of pre−mRNA in plants [56]. In darkness, phytochrome phyB resides in the cytoplasm as an inactive red light−absorbing form (phyB−Pr) [104]. Exposure to red light induces phyB to transition into its active far−red light−absorbing form (phyB−Pfr), which then relocates to the nucleus to form photobodies. The PrLP SWAP1 (SUPPRESSOR OF WHITE APRICOT/SURP RNA−BINDING DOMAIN CONTAINING PROTEIN1) interacts with the splicing factors SFPS (SPLICING FACTOR FOR PHYTOCHROME SIGNALING) and RRC1 (REDUCED RED–LIGHT RESPONSES IN CRY1CRY2 BACKGROUND1) to form the SFPS–RRC1–SWAP1 complex for pre–mRNA splicing. This complex is recruited by photobodies and interacts with the spliceosome to mediate the alternative splicing of pre−mRNA [56,105,106] (Figure 2D).

### 4.3. Reproductive Growth

Plant reproductive growth, including flowering and fruiting, represents a crucial stage in the life span of flowering plants. Flowering time is tightly regulated by both genetic factors and environmental signals, with at least six pathways involved in this regulatory network [107,108]. FT (FLOWERING LOCUS T), a major component of florigen, controls the transition from vegetative to reproductive growth [109,110]. Additionally, FLC (FLOWERING LOCUS C), a MADS–box transcription factor, is a critical inhibitor of *FT* transcription in the autonomous and vernalization pathways [111,112]. The expression of FLC is regulated at the epigenetic, transcriptional, and post–transcriptional levels, with PrLPs reported to participate in these processes [111,113,114].

Under warm conditions, the activation of FLC expression is controlled by the trimethylation of histone H3 at lysine 36 and lysine 4 (H3K36me3 and H3K4me3), which depends on the transcriptional activator FRIGIDA (FRI) complex [115,116]. This complex is composed of FRI, FRL1 (FRIGIDA–LIKE 1), and several chromatin modifiers and transcriptional co–activators. In cold temperatures, FRI and FRL1 form nuclear condensates that sequester FRI away from the *FLC* promoter. The PrLP FRL1 participates in the formation of these condensates, and the disruption of the FRI condensate occurs in the *frl1*–*1* mutant [117] (Figure 3(Aa)). LD (LUMINIDEPENDENS) interacts with SDG26 (SET DOMAIN GROUP 26) and the H3K4 demethylase FLD (FLOWERING LOCUS D) to form the FLD/LD/SDG26 complex. The complex acts as a negative regulator of FLC expression, thus inhibiting the accumulation of H3K4me1 within the *FLC* gene body [118] (Figure 3(Ab)). As a PrLP, LD can form condensates in the nucleus [14]. However, whether its phase separation is involved in the assembly of the FLD/LD/SDG26 complex requires further investigation. Additionally, the MADS–box transcription factor FLM (FLOWERING LOCUS M) inhibits *FT* transcription to prevent early flowering [119,120]. The transcription of *FLM* is regulated by the PrLP UBA2c. UBA2c can bind directly to *FLM* chromatin and inhibit the trimethylation of H3K27, a marker associated with transcriptional repression [121]. Interestingly, UBA2c forms gel–like speckles in the nucleus, dependent on its PrLD [121]. This function cannot be substituted by the PrLD of FUS, which is known to drive LLPS, suggesting that the gel–like condensates formed by the UBA2c PrLD domain play a distinct role in *Arabidopsis* [121] (Figure 3(Ac)).

At the transcriptional level, two PrLPs, the P–body component DCP5 and the floral repressor SSF (SISTER OF FCA), work together to regulate the transcription of FLC. SSF serves as a “scaffold” protein anchored to the *FLC* locus, facilitating the recruitment of DCP5 to the FLC chromatin, where it undergoes LLPS. This LLPS depends on the presence of the PrLD in both SSF and DCP5. The formation of these nuclear condensates is essential for the repression of *FLC* transcription. Lacking the PrLDs in either DCP5 or SSF leads to failure in repressing *FLC* transcription and an inability to rescue the flowering time phenotype [45] (Figure 3(Ad)). Additionally, the PrL–RBP HRLP suppresses *FLC* transcription by promoting R–loop formation near the intron of *FLC*, resulting in reduced recruitment of RNA Pol II. HRLP forms a complex with the splicing factor SR45 and undergoes LLPS to form nuclear condensates. These condensates are necessary for inhibiting the co–transcriptional splicing and transcription of *FLC* (Figure 3(Ae)). Notably, the LLPS of HRLP is mediated not by its C–terminal PrLD but by IDRs in its N–terminal region [47].

In addition, PrLPs also regulate post–transcription processes to affect flowering. The splicing efficiency of the *FLC* pre–mRNA is a critical determinant in producing mature *FLC* mRNA [54]. Two PrL–RBPs, KHZ1 (CCCH zinc–finger and KH domain–containing proteins 1) and KHZ2, can form homodimers or heterodimers. Their accumulation in nuclear speckles is associated with the reduced splicing efficiency of the *FLC* pre–mRNA [54] (Figure 3(Af)). *COOLAIR*, the long antisense RNAs transcribed from the downstream region of the *FLC* locus, are key modulators of the chromatin environment at the *FLC* locus [122,123]. The PrL–RBP FCA can form nuclear condensates in collaboration with the polymerase and nuclease components of the RNA 3’–end processing machinery and FLL2, a PrLP with coiled–coil domains. These condensates regulate polyadenylation at specific genomic sites, including proximal sites of *COOLAIR* nascent transcripts, thereby reducing transcriptional read–through. FLL2 promotes the formation of these condensates, and its overexpression leads to an increase in the size and number of these condensates. Other PrLD–containing proteins, such as FLL1, FPA, and FY, have been identified within the FCA condensates, indicating a complex interplay among these factors in gene expression regulation through phase–separated nuclear condensates [52] (Figure 3(Ag)). Moreover, PrLPs are involved in regulating mRNA stability through the recognition of m^6^A–modified mRNAs. In *Arabidopsis*, CPSF30–L enhances the formation of liquid–like nuclear bodies, primarily recognizing m^6^A–modified far–upstream elements (FUEs), and is crucial for controlling polyadenylation site choice by recruiting the polyadenylation complex. The *cpsf30–l* mutant leads to an elongation of the 3′ untranslated region (3′ UTR) of specific transcripts, including the floral transition–related gene SOC1 (SUPPRESSOR OF OVEREXPRESSION OF CONSTANS1), resulting in delayed flowering [51] (Figure 3(Ah)). Similarly, in rice, the PrLP EHD6 (EARLY HEADING DATE6) recruits YTH07, a PrLP recognized as an m^6^A reader, to form RNP granules. These granules facilitate the sequestration of the mRNA of the flowering repressor *OsCOL4* (*CONSTANS*–*like 4*) in rice, which is critical for reducing OsCOL4 abundance and accelerating flowering [124] (Figure 3C).

Inflorescence formation is a prerequisite for flowering. In tomatoes, the PrLP ALOG transcription factor family plays a significant role in both the regulation of flowering time and the development of inflorescence. The TMF (TERMINATING FLOWER) and its paralogous TFAMs (TMF FAMILY MEMBERs) form heterotypic transcriptional condensates. These condensates collaboratively repress the expression of the floral identity gene AN (ANANTHA). This repression fine–tunes the transition of SAM maturation and the development of inflorescences. Loss–of–function mutations in single or multiple combinations of TMF and TFAMs result in varying degrees of early flowering and alterations in inflorescence structure [125] (Figure 3D).

Prezygotic interspecific incompatibility prevents the formation of unfavorable hybrids between species [126,127,128]. The PrLP SPRI2 (STIGMATIC PRIVACY 2), a member of the SHI (SHORT–INTERNODES) transcription factor family, along with its paralogue SRS5 (SPRI2–like 5), is crucial for this process. SPRI2 and SRS5 are essential for the rejection of pollen from other species on pistils by activating the transcription of genes related to cell wall modification. Notably, SPRI2 localizes within nuclear condensates through LLPS, which is governed by its PrLD [128] (Figure 3B).

## 5. PrLPs Involved in Environmental Perception

After sensing changes in both intracellular and extracellular environments, including pH, ion concentration, and temperature, PrLPs can rapidly form condensates via phase separation [129,130,131]. These condensates are typically dynamic and reversible, enabling precise regulation of intracellular processes when the environment changes.

### 5.1. Abiotic Stress

Hyperosmotic stress causes cellular dehydration, which diminishes cell volume and subsequently increases intracellular macromolecular crowding. This increased crowding enhances the likelihood of molecular interactions, potentially serving as a cellular sensing mechanism to detect hyperosmotic stress. In *Arabidopsis*, the transcriptional regulator SEUSS (SEU) has been identified as a sensor of molecular crowding. SEU contains an N–terminal PrLD (also referred to as IDR1), which enables SEU to form condensates under hyperosmotic conditions. SEU promotes the upregulation of genes involved in osmotic adjustment, thereby enhancing the plant’s resilience to hyperosmotic stress [132] (Figure 4(Aa)).

Seed maturation involves an essential dehydration phase that protects embryos from external stresses and maintains dormancy [133]. Under favorable conditions, seeds rapidly absorb water, hydrate, and resume cellular activities, leading to germination. FLOE1, a key water–sensing PrLP in seeds, can undergo reversible hydration–dependent phase separation. This process results in the formation of cytoplasmic condensates during seed imbibition, which is essential for the embryo to detect water availability and initiate germination under optimal conditions. Under salt or hyperosmotic stresses, FLOE1 may suppress germination by reducing condensate formation, thereby enhancing the plant’s survival in adverse environments. Notably, the seeds of *floe1*–*1* mutants exhibit increased germination rates under salt and hyperosmotic stress. FLOE1 contains two IDRs: the N–terminal DS domain rich in aspartic acid (D) and serine (S), and the C–terminal QPS domain, which is a PrLD enriched in glutamine (Q), proline (P), and serine (S). Deleting the QPS domain prevents phase separation, resulting in a loss–of–function phenotype. However, deleting the DS domain results in a gel–like condensate, exhibiting a gain–of–function phenotype that promotes germination under stress. Furthermore, natural variation in FLOE1 suggests that different isoforms may modulate germination timing in natural populations, aligning with diverse climatic adaptations [134,135] (Figure 4(Ab)).

The transcription factor STM plays a pivotal role in sensing salt stress. During salt stress, STM forms condensates, then recruits the Mediator complex subunit MED8 and BELLs proteins, including ATH1 (ARABIDOPSIS THALIANA HOMEOBOX 1), RPL (REPLUMLESS, also known as PENNYWISE and BELLRINGER), and PNF (POUNDFOOLISH). Upon exposure to elevated salt stress, plant cells exhibit enhanced STM/BELLs/MED8 condensate formation. These condensates amplify the STM’s transcriptional activity and stimulate the expression of downstream target genes, thereby promoting salt tolerance and increased shoot branching. Notably, the proteins within this complex are all classified as PrLPs, with RPL likely playing a key role in condensate formation [43] (Figure 4(Ac)).

RNA m^6^A modification is a crucial epigenetic modification that regulates various aspects of plant development and stress responses [136,137,138]. The ECT8, an m^6^A reader with a PrLD, is involved in regulating mRNA stability under salt stress. Both transcriptional and translational levels of ECT8 are upregulated in response to salt stress. ECT8 directly interacts with decapping protein DCP5 and undergoes LLPS to form PBs, which are crucial for the degradation of m^6^A–modified RNAs that encode negative regulators of the salt stress response [139] (Figure 4(Ad)).

Heat stress impedes normal developmental processes and reduces plant productivity [140,141]. To mitigate the effects of heat, plants activate heat stress transcription factors (HSFs), which respond to elevated temperatures by initiating the transcription of heat–responsive genes [142,143]. In *Arabidopsis*, ALBA proteins, specifically the PrLPs ALBA4 and ALBA6, undergo LLPS under heat stress, leading to the formation of PBs or SGs. These cytoplasmic granules interact with HSF mRNAs and other transcripts, stabilizing them under high–temperature conditions and ensuring their translation when conditions return to normal [55]. Additionally, OLIGOURIDYLATE BINDING PROTEIN UBP1b, a PrLP and another component of SGs, serves a similar function. UBP1b can bind mRNAs and undergo LLPS during heat stress to form SGs, thus protecting heat stress–related mRNAs from degradation [144]. Concurrently, PrLP ARGONAUTE1 (AGO1) forms condensates and accumulates within SGs under heat stress. The sequestration of AGO1 within SGs protects AGO1 and ensures that the RNA–silencing activity of AGO1 remains unaffected by the stress [145] (Figure 4(Ae)).

Elevated temperatures prompt plants to flower prematurely, an adaptive strategy that ensures the timely completion of their reproductive cycle [146,147]. Under normal conditions, the evening complex, comprising ELF3 (EARLY FLOWERING 3), ELF4, and LUX (LUX ARRHYTHMO), binds to the promoter regions of flowering–associated genes and inhibits their transcription [42,148,149]. Under heat stress, the PrLP ELF3 undergoes LLPS to form ELF3 condensates, which sequester ELF3 and prevent its assembly with ELF4 and LUX within the evening complex. As a result, the transcriptional repression of flowering genes is alleviated, accelerating the transition to the flowering stage [42]. ELF3 represents a novel thermosensory mechanism, allowing plants to dynamically adjust their growth and developmental programs in response to environmental stimuli (Figure 4(Af)).

### 5.2. Biotic Stress

During growth and development, plants are frequently attacked by various pathogens and insects. To combat these biotic stresses, plants have developed various defense mechanisms, including the production of defense–related hormones and secondary metabolites and rapid gene expression regulation [150]. Salicylic acid (SA) is a key defense–related hormone that stimulates the expression of stress–responsive genes to promote immune responses [151,152,153]. However, plants must balance an overly robust immune response to avoid irreversible cellular damage [154,155]. ECT1, an m^6^A reader with a PrLD at its N–terminus, is a key regulator in this process. ECT1 binds to m^6^A–modified SA–induced mRNAs and undergoes LLPS to form condensates. These condensates may transform into PBs or SGs, which are involved in mRNA degradation or storage, thereby reducing mRNA levels and negatively regulating the immune response [156] (Figure 4(Ba)). Additionally, ECT1 interacts with its homolog ECT9 to undergo phase separation, modulating immune responses to non–pathogenic stimuli and preventing an overactive immune system [157]. Components of PBs, including DCP1, DCP2, and XRN4, participate in the mRNA decay process triggered by microbe–associated molecular patterns (MAMPs). MAMP–activated MAP kinases (MPK3 and MPK6) phosphorylate DCP1, enhancing its interaction with XRN4, a 5′–3′ RNA exonuclease responsible for the degradation of decapped mRNAs [74] (Figure 4(Bb)).

Nucleotide–binding leucine–rich repeat receptors (NLRs) are pivotal intracellular immune receptors for disease resistance [158]. However, the expression of NLRs under normal conditions may trigger autoimmunity, requiring tight regulation of their transcription [159]. The PrL–RBP FPA is involved in this regulatory mechanism. FPA can interact with RNA cleavage and polyadenylation factors, leading to the premature cleavage and polyadenylation of NLR transcripts. This process controls the functional expression of NLRs and affects immunity [160]. Although FPA contains a PrLD, its capacity to undergo LLPS and form condensates during RNA processing remains to be determined.

RNA interference (RNAi) is a crucial mechanism for effecting the expression of immune–related genes in response to pathogens and viruses [161,162]. Several PrLPs are involved in small RNA biosynthesis and gene silencing. D–bodies, a type of nuclear RNP granule, are crucial for microRNA (miRNA) biosynthesis and processing [57]. DEAD–box RNA helicases, including PrLPs RH6, RH8, and RH12, interact with SERRATE (SE), a key component of D–bodies, and promote LLPS to form D–bodies. Upon Turnip mosaic virus (TuMV) infection, the accumulation of these helicases in the nucleus decreases, leading to a reduction in D–body levels. This reduction correlates with the negative regulation of TuMV infections [53] (Figure 4(Bc)). Furthermore, the PrLP SGS3 undergoes LLPS and recruits RDR6 to form the SGS3/RDR6 body (also known as the siRNA body) [78,163]. The SGS3/RDR6 body modulates the production of siRNAs, thereby facilitating plant defense against viral infection [163]. Additionally, the core components of the RNA–induced silencing complex (RISC), known as ARGONAUTE proteins (AGO1, AGO2, AGO3, and AGO5), contain PrLDs and are potential phase–separating proteins [78,164].

Plants produce unique biochemical substances known as allelopathic compounds, which inhibit the normal development of neighboring plants in the competitive struggle for survival [165]. Phenolic acid (PA), commonly secreted by plant roots, can enter the plant cell nucleus. The PrLP RBP47B acts as a PA receptor. Upon binding to PAs, RBP47B undergoes LLPS, forming SGs. These SGs incorporate mRNA, ribosomes, rRNA, and various other cellular components, leading to the comprehensive suppression of translation in plant cells affected by allelopathic substances [166] (Figure 4(Bd)).

## 6. Conclusions and Future Prospects

In this review, we focus on the structural characteristics of PrLPs and their crucial roles in regulating phase separation. We also discuss the molecular functions of PrLPs and their involvement in various biological processes, highlighting their broad impact on cellular function. While most research has been conducted using the model organism *Arabidopsis*, our understanding of PrLPs in other plant species remains limited. Further studies could shed light on PrLP functions across the plant kingdom. Recent findings have also underscored the interaction between MLOs and cellular membranes, which may influence membrane properties and processes [80]. Although PrLPs are enriched in many membrane–associated processes, their specific roles in these processes remain largely unexplored. Elucidating these functions could significantly enhance our understanding of cellular dynamics. Additionally, investigating PrLPs from an evolutionary perspective could uncover their new roles in environmental adaptations and evolutionary processes. Such an approach may reveal conserved and complex mechanisms, thereby deepening our understanding of the evolutionary significance of PrLPs.

## Figures and Tables

**Figure 1 plants-13-02666-f001:**
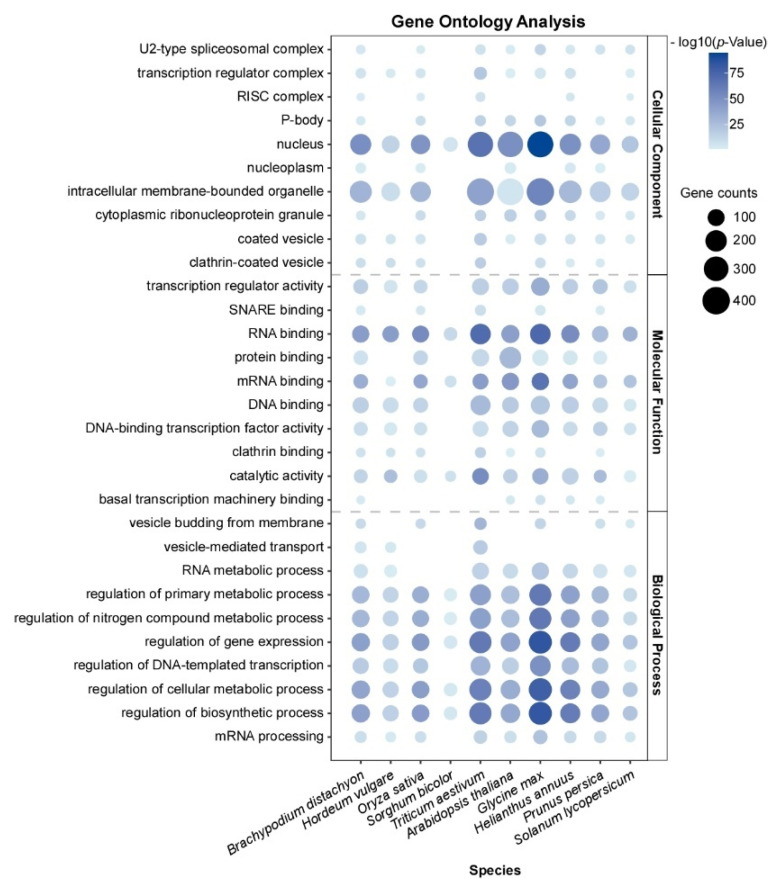
Gene Ontology (GO) analysis of prion–like protein (PrLP) genes across 10 plant species (Brachypodium distachyon, Hordeum vulgare, Oryza sativa, Sorghum bicolor, Triticum aestivum, Arabidopsis thaliana, Glycine max, Helianthus annuus, Prunus persica, and Solanum lycopersicum). Bubble size represents the number of genes, while color variation represents different *p*–values.

**Figure 2 plants-13-02666-f002:**
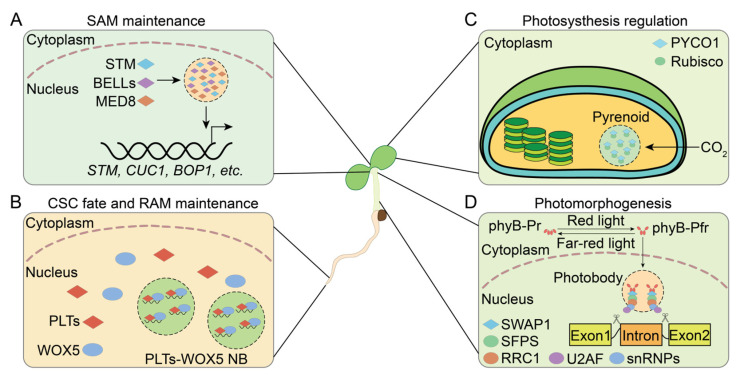
Prion−like proteins (PrLPs) regulate plant meristem maintenance and light signaling. (**A**) STM, BELLs, and MED8 form condensates to maintain the SAM. (**B**) PLTs form NBs with WOX5 and RNA to control the destiny of the CSC and sustain the RAM. (**C**) PYCO1 forms pyrenoids to bind rubisco, enhancing the efficiency of CO_2_ fixation in photosynthesis. (**D**) The SWAP/SFPS/RRC1 complex interacts with photobodies to regulate alternative splicing of pre–mRNAs. Diamonds indicate PrLPs; ellipses represent other proteins; dashed circles indicate droplet–like condensates; solid–line circles represent gel–like condensates. BELL: BEL1−like; CSC: columella stem cell; NB: nuclear body; Pfr: physiological active far−red form; phyB: phytochrome B; PLT: PLETHORA; Pr: physiological inactive red form; PYCO1: pyrenoid component 1; RAM: root apical meristem; RRC1: REDUCED RED−LIGHT RESPONSES IN CRY1CRY2 BACKGROUND1; SAM: shoot apical meristem; SFPS: SPLICING FACTOR FOR PHYTOCHROME SIGNALING; snRNP: small nuclear ribonucleoproteins; STM: SHOOT MERISTEMLESS; SWAP1: SUPPRESSOR OF WHITE APRICOT/SURP RNA−BINDING DOMAIN CONTAINING PROTEIN1; WOX5: WUSCHEL−RELATED HOMEOBOX 5.

**Figure 3 plants-13-02666-f003:**
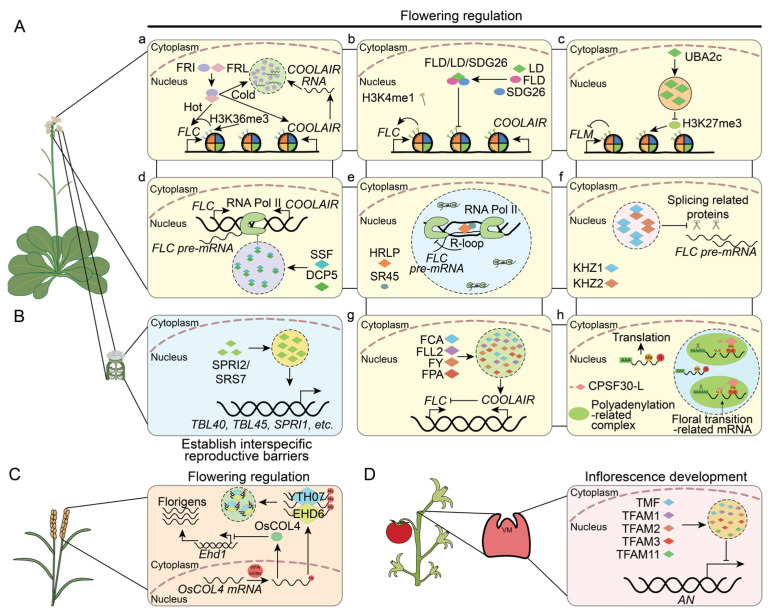
Prion–like proteins (PrLPs) regulate plant reproductive growth. (**A**) PrLPs regulate flowering and fruit development. (**B**) SPRI2/SRS7 condensates facilitate the development of interspecific reproductive barriers. (**C**) YTH07 and EHD6 bind to m^6^A–modified mRNA and form condensates to initiate flowering in rice. (**D**) TMF and TFAM1/2/3/11 form condensates to modulate the development of inflorescences in tomatoes. Diamonds indicate PrLPs; ellipses represent other proteins; dashed circles indicate droplet–like condensates; solid–line circles represent gel–like condensates. AN: ANANTHA; DCP5: DECAPPING 5; EHD6: EARLY HEADING DATE 6; FCA: FLOWERING CONTROL LOCUS A; FLC: FLOWERING LOCUS C; FLD: FLOWERING LOCUS D; FLM: FLOWERING LOCUS M; FRI: FRIGIDA; FRL: FRIGIDA like; FT: FLOWERING LOCUS T; H3K4me1: monomethylated H3K4; H3K27me3: trimethylated H3K27; H3K36me3: trimethylated H3K36; HRLP: hnRNP R–LIKE PROTEIN; LD: LUMINIDEPENDENS; m^6^A: N^6^–methyladenosine; OsCOL4: CONSTANS–like 4; RNA Pol II: RNA polymerase II; SDG26: SET DOMAIN GROUP 26; SPRI2: STIGMATIC PRIVACY 2; SR45: SERINE/ARGININE–RICH 45; SSF: SISTER OF FCA; TFAM: TMF family member; TMF: TERMINATING FLOWER.

**Figure 4 plants-13-02666-f004:**
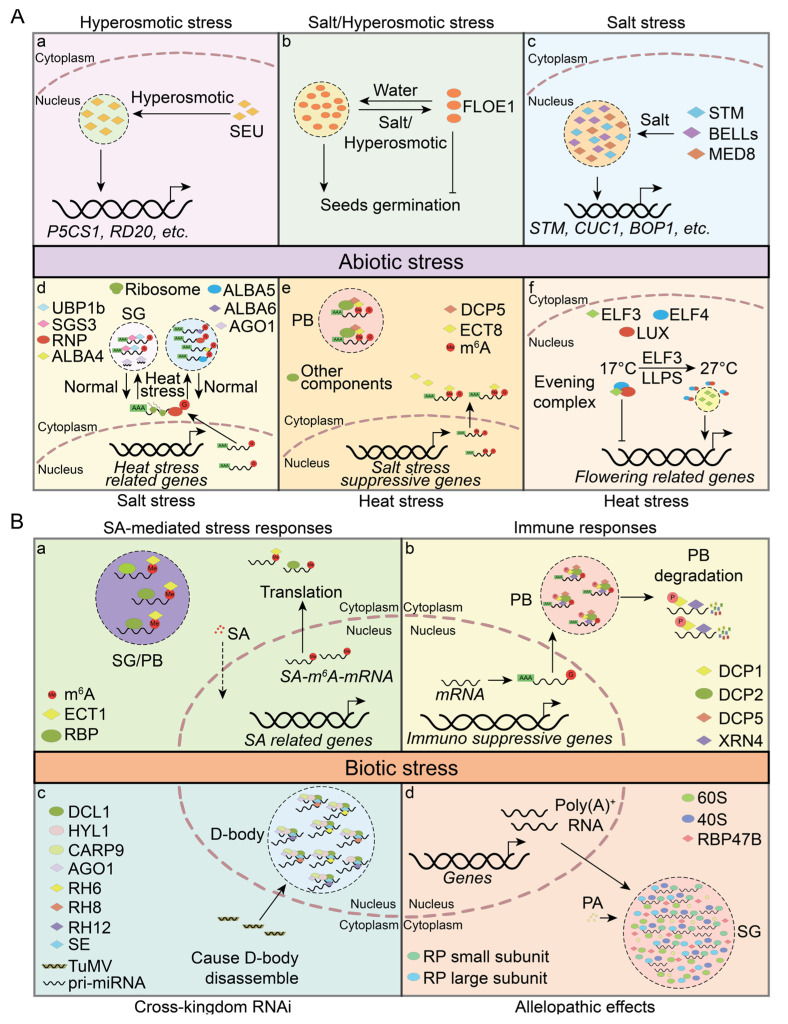
PrLPs regulate plant adaptation to stresses. (**A**) PrLPs coordinate the plant’s response to abiotic stress. (**a**,**b**). PrLPs modulate plant responses to hyperosmotic stress. (**b**–**d**). PrLPs modulate plant responses to salt stress. (**e**,**f**). PrLPs govern plant responses to heat stress. (**B**) PrLPs exhibit responses to biotic stressors. (**a**,**b**). SGs or PB for immune responses. (**c**). D–bodies involved in pri–miRNA processing. (**d**). SGs assemble in response to allelopathic effects. Diamonds indicate PrLPs; ellipses represent other proteins; dashed circles indicate droplet–like condensates. AGO1: ARGONAUTE1; BELL: BEL1–like; CARP9: CONSTITUTIVE ALTERATIONS IN THE SMALL RNA PATHWAYS9; D–body: dicing body; DCL1: Dicer–like1; DCP: DECAPPING; ECT: EVOLUTIONARILY CONSERVED C–TERMINAL REGION; ELF3: EARLY FLOWERING 3; ELF4: EARLY FLOWERING 4; HYL1: Hyponastic Leaves1; m6A: N6–methyladenosine; PA: phenolic acid; PB: processing body; pri–miRNA: primary microRNA; RBP: RNA–binding protein; RBP47B: RNA–binding protein 47B; RH: RNA helicase; RNAi: RNA interference; RNP: ribonucleoprotein; RP: ribosome protein; SA: salicylic acid; SE: SERRATE; SEU: SEUSS; SG: stress granule; SGS3: SUPPRESSOR OF GENE SILENCING 3; STM: SHOOT MERISTEMLESS; TuMV: Turnip mosaic virus; UBP1b: OLIGOURIDYLATE BINDING PROTEIN 1b; VCS: VARICOSE; XRN: 5′–to–3′ exonuclease exoribonuclease.

## Data Availability

The data presented in this study are available in Appendix A.

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
