# Peer review of "Prion–like Proteins in Plants: Key Regulators of Development and Environmental Adaptation via Phase Separation"

_plants, 2024, doi:10.3390/plants13182666_

Round 1

Reviewer 1 Report

Comments and Suggestions for Authors

The submitted manuscript addresses an intriguing subject, which, over the past years has been given particular attention to due to obvious reasons: understanding the nature and unravelling the biology of PrLPs is of pivotal importance in order to be able to monitor its presence and conformation so as to bring endeavours aimed at controlling its dynamics to ultimate success.

The ms. is well-written and presented in a concise and perspicuous manner. Especially the highlighting of the increasing appreciation of the importance of PrLPs present in living systems is worth mentioning.

Below, I give an overall evaluation of the text underlining those points at which, I think, the ms. would profit from being altered:

-Line 33: the paragraph is in contrast to the previous one, more precisely as it summarises the basic biology of prions and in this paragraph the authors mention phase separation, which, as a term is expounded in the text that follows, but for those, who are not familiar with the biochemistry of prions it should be emphasised in a more consistent, contextual way why it is mentioned (at this point of the introductory part).

-Further: it appears a bit confusing that ‘Introduction’ seems to be a subheading as well as ’Structural features…’ (Line 45), although the latter should be categorised as one of the points mentioned under the introduction.

-Line 45-93: The expounding of the concept of the review article is easy to follow, but the sentences should have a more pronounced contextual relationship among them; in its present form the paragraph appears to delineate the core issue as to what is known about the structure and features of PrLPs in a bit ‘lumped’ way.   

- From Line 94 onwards: it should perhaps be advisable to say a few words about what is known about PrLPs in general (and/or in the animal kingdom) before striking up the plant kingdom

-Line 116: ‘..formation can sequester…’ should be rephrased (formation itself does not ‘sequester’)

-Line 124: instead of the singular ‘forms’ plural of the verb (form) may be considered

-Line 137: if the authors refer to the process (of condensates) then ‘enhance’ should be ‘enhances’

-Line 148: ‘Cajal bodies’ should be expounded

Line 204: after LLPS space should be inserted

-Line 207: likewise (after the word: ‘differentiation’)

-Line 216: the same space issue (the authors should carefully check this throughout the whole text

-Line 239: ‘2’ should be low index (in carbon dioxide)

-Line 341-342: Figure 2 is informative and complex, but seems to be too busy (perhaps a division of the captions depicting the different aspects of the role played by PrLPs may be considered)

-Line 500-538: would not it be advisable to swap point 6. and 7.? (A reader would expect the conclusion and the mentioning of future prospects (instead of ‘perspectives’) to summarise a (review) paper.

In conclusion: the authors give a comprehensive review on the role PrLPs play in crucial developmental processes of plants and in their combatting both abiotic and biotic stress conditions, which can be deemed to be an informative summary of the ‘state-of-the-art’ of endeavours attempted to disentangle the dynamic role these proteins play in regulatory and stress mechanisms.

I believe that the eventual publication of the work could contribute to the gathering of the knowledge of the plant scientists’ community in this field and also stimulate ongoing and to-be-launched research initiatives in the future.

Upon the consideration of the minor modifications suggested above, I recommend the acceptance of the manuscript.  

Comments on the Quality of English Language

The overalll English is fine, only minor modifications are sugggested.

Author Response

We would like to sincerely thank you for your thorough and insightful review of our manuscript. We are grateful for the specific suggestions you provided to enhance its structure and readability. Below we address each of your comments in detail:

Comment 1: -Line 33: the paragraph is in contrast to the previous one, more precisely as it summarizes the basic biology of prions and in this paragraph the authors mention phase separation, which, as a term is expounded in the text that follows, but for those, who are not familiar with the biochemistry of prions it should be emphasized in a more consistent, contextual way why it is mentioned (at this point of the introductory part).

Response 1: We agree that the transition from the basic biology of prions to the concept of phase separation could be clearer. To address this, we have added a transitional sentence in the revised manuscript (Line 34-36).

Comment 2: -Further: it appears a bit confusing that ‘Introduction’ seems to be a subheading as well as ’Structural features…’ (Line 45), although the latter should be categorized as one of the points mentioned under the introduction.

Response 2: We agree that clarity in section headings is important. However, we consider that placing the “Structural Features and Mechanisms” section under the "Introduction" would result in an overly lengthy introductory section. Therefore, “Structural Features and Mechanisms” is introduced as a major section under the main body of the manuscript.

Comment 3: -Line 45-93: The expounding of the concept of the review article is easy to follow, but the sentences should have a more pronounced contextual relationship among them; in its present form the paragraph appears to delineate the core issue as to what is known about the structure and features of PrLPs in a bit ‘lumped’ way.

Response 3: We appreciate the reviewer’s suggestion. We have reorganized and reworded several sentences to improve the logical flow between concepts (line 55-100).

Comment 4: - From Line 94 onwards: it should perhaps be advisable to say a few words about what is known about PrLPs in general (and/or in the animal kingdom) before striking up the plant kingdom.

Response 4: Thanks for your suggestion and we have revised the beginning of the paragraph (line102-104).

Comment 5: -Line 116: ‘..formation can sequester…’ should be rephrased (formation itself does not ‘sequester’).

Response 5: We agree with the reviewer’s suggestion and have reworded the sentence (line 124-125).

Comment 6: -Line 124: instead of the singular ‘forms’ plural of the verb (form) may be considered

Response 6: We thank the reviewer’s suggestion and have corrected the verb "forms" to the plural "form" in the revised manuscript (line 133)

Comment 7: -Line 148: ‘Cajal bodies’ should be expounded

Response 7: We have described the function of Cajal bodies in line 160-161.

Comment 8: -Line 204: after LLPS space should be inserted

Comment 9:-Line 207: likewise (after the word: ‘differentiation’)

Comment 10:-Line 216: the same space issue (the authors should carefully check this throughout the whole text

Response 8: We have carefully reviewed the manuscript to correct all spacing issues.

Comment 11: -Line 239: ‘2’ should be low index (in carbon dioxide)

Response 9: The “2” in “CO2” has been corrected to a subscript (line 243).

Comment 12: -Line 341-342: Figure 2 is informative and complex, but seems to be too busy (perhaps a division of the captions depicting the different aspects of the role played by PrLPs may be considered)

Response 10: We thank your suggestion and we have split Figure 2 into two separate figures (now labeled Figure 2 and Figure 3) to reduce complexity and enhance clarity.

Comment 13: -Line 500-538: would not it be advisable to swap point 6. and 7.? (A reader would expect the conclusion and the mentioning of future prospects (instead of ‘perspectives’) to summaries a (review) paper.

Response 11: We agree with your suggestion. We have renamed subheading 6 to “Conclusions and Future Prospects”, and moved the content of section 7 to the Appendix A.

Reviewer 2 Report

Comments and Suggestions for Authors

The manuscript titled "Prion-like Proteins in Plants: Key Regulators of Development and Environmental Adaptation via Phase Separation" provides a comprehensive and insightful review of prion-like domains (PrLDs) in plants, focusing on their structural features and their critical roles in liquid-liquid phase separation (LLPS). This review offers valuable contributions to the current understanding of the physiological and functional roles of prion-like proteins (PrLPs) in plant development and responses to environmental stress.

The manuscript is well-structured, guiding the reader logically through the various aspects of PrLDs, from their basic structural characteristics to their functional roles in cellular processes. One of the strongest points of the work is its extensive literature review, which not only covers existing knowledge but also incorporates recent findings on the cellular localization and potential functions of PrLPs across various plant species. The gene ontology (GO) analysis conducted by the authors provides additional depth, highlighting the molecular functions and diverse roles of PrLPs.

Overall, this review is a significant contribution to the field of plant biology and protein phase separation, offering an in-depth examination of PrLPs in plant physiology.  I recommend the manuscript for publication after minor revisions.

There are a few minor points that could enhance the manuscript:

The citations within the text should be revised to ensure consistent formatting, specifically ensuring that spaces are placed between the text and citation brackets (e.g., "[80]").

Author Response

We would like to sincerely thank you for your thorough and insightful review of our manuscript. We are grateful for the specific suggestions you provided to enhance its structure and readability. Below we address each of your comments in detail:

Comment 1: The citations within the text should be revised to ensure consistent formatting, specifically ensuring that spaces are placed between the text and citation brackets (e.g., "[80]").

Response 1: We have carefully reviewed the manuscript to correct all spacing issues.